# A Comprehensive Library for Benchmarking Multi-class Visual Anomaly Detection

## Abstract

Visual anomaly detection aims to identify anomalous regions in images through unsupervised learning paradigms, with increasing application demand and value in fields such as industrial inspection and medical lesion detection. Despite significant progress in recent years, there is a lack of comprehensive benchmarks to adequately evaluate the performance of various mainstream methods across different datasets under the practical multi-class setting. The absence of standardized experimental setups can lead to potential biases in training epochs, resolution, and metric results, resulting in erroneous conclusions. This paper addresses this issue by proposing a comprehensive visual anomaly detection benchmark, *ADer*, which is a modular framework that is highly extensible for new methods. The benchmark includes multiple datasets from industrial and medical domains, implementing fifteen state-of-the-art methods and nine comprehensive metrics. Additionally, we have proposed the GPU-assisted *ADEval* package to address the slow evaluation problem of metrics like time-consuming mAU-PRO on large-scale data, significantly reducing evaluation time by more than *1000-fold*. Through extensive experimental results, we objectively reveal the strengths and weaknesses of different methods and provide insights into the challenges and future directions of multi-class visual anomaly detection. We hope that *ADer* will become a valuable resource for researchers and practitioners in the field, promoting the development of more robust and generalizable anomaly detection systems. Full codes have been attached in Appendix and will be open-sourced.

## 1 Introduction

In recent years, with the rapid advancement in model iteration and computational power, Visual Anomaly Detection (VAD) has made significant progress across various fields Liu et al. (2024); Cao et al. (2024). It has become a crucial component in key tasks such as industrial quality inspection and medical lesion detection. Due to its unsupervised experimental setup, VAD demonstrates immense application value in real-world scenarios where the yield rate is high, defect samples are difficult to obtain, and potential defect patterns are diverse. However, the field faces challenges such as small dataset sizes and insufficient evaluation metrics, resulting in potentially unfair comparison outcomes due to differing training recipes among methods. Moreover, most methods have not been compared on the latest large-scale datasets (*e.g.*, Real-IAD Wang et al. (2024) and COCO-AD Zhang et al. (2024a)) and new evaluation metrics (*e.g.*, mAD Zhang et al. (2023a) and mIoU-max Zhang et al. (2024a)). The fundamental issue lies in the absence of standardized training strategies, akin to those in object detection, to evaluate different algorithms. Factors such as training epoch and resolution can potentially affect evaluation results, leading to erroneous conclusions.

To address this pressing issue, we believe that establishing a comprehensive and fair benchmark is crucial for the sustained and healthy development of this field. Therefore, we have constructed an integrated *ADer* library, benchmarking state-of-the-art methods by utilizing a unified evaluation interface under the more practical multi-class setting. This library is designed as a highly extensible modular framework (see Sec. 3), allowing for the easy implementation of new methods. Specifically, the framework integrates multiple datasets from industrial, medical, and general-purpose domains (see Sec. 3.2), and implements fifteen state-of-the-art methods (including augmentation-based, embedding-based, reconstruction-based, and hybrid methods, see Sec. 3.1) and nine comprehensive evaluation metrics (see Sec. 3.3), ensuring thorough and unbiased performance evaluation for each method.

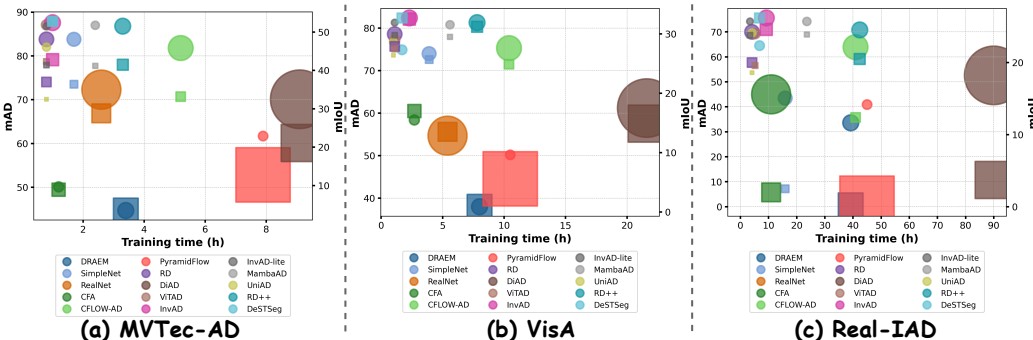

Figure 1: Intuitive benchmarked results comparison on MVTec AD Bergmann et al. (2019) (Left), VisA Zou et al. (2022) (Middle), and Real-IAD Wang et al. (2024) (Right) datasets among mainstream methods. For each dataset, the horizontal axis represents the training time for different methods, the left vertical axis represents mAD Zhang et al. (2023a) (marked as circles, with radius indicating model parameter count), and the right vertical axis represents mIoU-max Zhang et al. (2024a) (marked as squares, with side length indicating model FLOPs).

Additionally, to address the efficiency issue of evaluating time-consuming metrics like mAU-PRO on large-scale data, we have developed and open-sourced the GPU-assisted ADEval package (see Sec. 3.6), significantly reducing evaluation time by over 1000 times, making previously impractical extensive detailed evaluations feasible on large-scale datasets.

Through extensive and fair experiments, we objectively reveal the strengths and weaknesses of different visual anomaly detection methods, comparing their efficiency (*i.e.*, model parameter count and FLOPs) and training resource consumption across different datasets, as shown in Fig. 1. Detailed results and analyses (see Sec. 4 and Appendix) elucidate the challenges of multi-class visual anomaly detection and provide valuable insights for future research directions.

In summary, the contributions of this paper are as follows: *1) Comprehensive benchmark*: We introduce a modular and extensible library termed *ADer* for visual anomaly detection, which implements and evaluates 15 state-of-the-art anomaly detection methods on 11 popular datasets with 9 comprehensive evaluation metrics. *2) GPU-assisted evaluation package*: We develop and will open-source the ADEval package for large-scale evaluation, significantly reducing the evaluation time of complex metrics by over 1000 times. *3) Extensive experimental analysis*: We conduct extensive experiments to objectively evaluate the performance of different methods, providing insights into their strengths, weaknesses, and potential areas for improvement. *4) Open-source resources*: We will open-source the complete ADer code, making it a valuable resource for the research community and promoting further advancements in the field.

## 2 BACKGROUND AND RELATED WORK

### 2.1 PROBLEM DEFINITION AND OBJECTIVE

Visual anomaly detection (VAD) is a critical task in computer vision, aiming at identifying patterns or instances in visual data that deviate significantly from the norm. These anomalies can manifest as industrial defects, medical lesion, or rare objects that are not typically present in the training data. The primary objective of VAD is to develop algorithms capable of discerning these irregularities with high accuracy and reliability. This task is particularly challenging due to the inherent variability and complexity of visual data, the scarcity of anomalous examples, and the need for robust generalization across diverse scenarios. In a formal context, multi-class VAD can be defined as follows: Given a training dataset $D_{train} = \{x_1, x_2, ..., x_n\}$ with $C$ categories and each visual image $x_i$ belonging to a specific category, the goal is to learn a unified AD model $M$ that can predict an anomaly score $s_i = M(x_i)$ for each image. This score reflects the likelihood of each pixel in $x_i$ being an anomaly. The model $M$ is typically trained on $D_{train}$ that predominantly contains normal instances, with the assumption that anomalies are rare and not well-represented in the training set. Inevitably, there are some mislabeled or inaccurately labeled noisy samples, which constitute inherent biases within the dataset. These are typically disregarded under standard settings. The performance of the model is

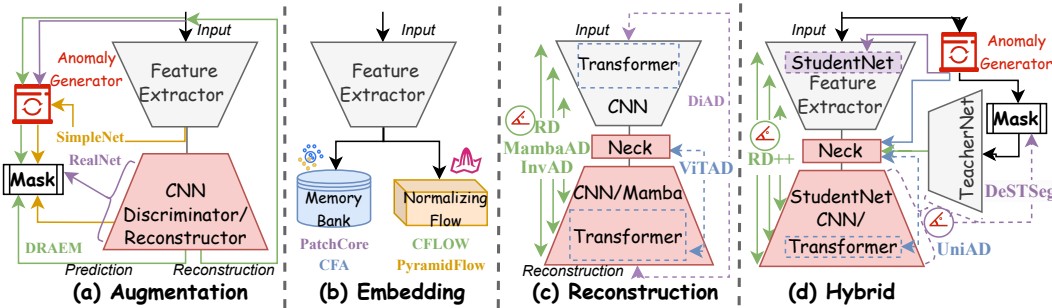

Figure 2: A comparative diagram of different frameworks for the benchmarked methods in Sec. 3.1.

then evaluated based on its ability to correctly identify anomalous images and their defect regions in a unified testset that contains normal and anomalous images.

## 2.2 CHALLENGES IN MULTI-CLASS VAD

The complexity of VAD arises from several factors: *1) Data Imbalance:* Anomalies are rare, leading to highly imbalanced datasets where normal instances and region areas vastly outnumber anomalous ones. *2) Variability of Anomalies:* Anomalies can vary widely in appearance, making it difficult to capture all possible variations during training. *3) Context Sensitivity:* The definition of what constitutes an anomaly can be context-dependent, requiring models to understand the broader context in which the visual data is situated. *4) Efficiency Requirements:* Many applications of VAD require real-time processing and limited GPU memory, necessitating efficient and scalable algorithms. *5) Comprehensive and Fair Evaluation:* Current methods exhibit significant differences in training hyperparameters and insufficient evaluation of performance metrics, so it is necessary to benchmark them using fair and standardized criteria. In this benchmark study, we systematically evaluate a range of state-of-the-art VAD methods (Sec. 3.1) across multiple datasets (Sec. 3.2) and comprehensive metrics (Sec. 3.3). Our goal is to provide a comprehensive assessment of current capabilities, identify key challenges, and suggest directions for future research in visual anomaly detection.

## 2.3 VISUAL ANOMALY DETECTION

Visual anomaly detection methods can generally be categorized into three types: *1) Augmentation-based methods* generate pseudo-supervised information for anomalies by creating abnormal regions Li et al. (2021); Zavrtanik et al. (2021), constructing anomalous data Zhang et al. (2021a); Hu et al. (2024), or adding feature perturbations Liu et al. (2023); Tien et al. (2023). This enables the model to learn the differences between normal and abnormal distributions. *2) Embedding-based methods* leverage pretrained models to extract powerful feature representations and judge anomalies in high-dimensional space. Typical approaches include distribution-map-based methods Gudovskiy et al. (2022); Lei et al. (2023), teacher-student frameworks Bergmann et al. (2020); Wang et al. (2021a), and memory-bank techniques Roth et al. (2022); Gu et al. (2023). *3) Reconstruction-based methods* use encoder-decoder architectures to locate anomalies by analyzing the reconstruction error. They typically include both image-level Akçay et al. (2019); Liang et al. (2023); He et al. (2024b) and feature-level approaches Deng & Li (2022); Zhang et al. (2023a; 2024a). There are also some hybrid methods You et al. (2022); Tien et al. (2023); Zhang et al. (2023c) that attempt to integrate multiple techniques to further enhance model performance.

**Basic network structures of VAD.** Early visual anomaly detection methods typically employ UNet-based autoencoder architectures Akcay et al. (2019); Liang et al. (2023); Zavrtanik et al. (2021). With advancements in foundational visual model structures He et al. (2016); Wang et al. (2021b); Liu et al. (2021); Zhang et al. (2021b; 2024b; 2023b) and pretraining techniques He et al. (2022); Caron et al. (2021), more recent methods often utilize models pretrained on ImageNet-1K Deng et al. (2009) as feature extractors, such as the ResNet He et al. (2016) series, Wide ResNet-50 Zagoruyko & Komodakis (2016), and EfficientNet-b4 Tan & Le (2019). Recently, benefiting from the dynamic modeling capabilities of Vision Transformers (ViT) Dosovitskiy et al. (2021), some studies De Nardin

Table 1: Attribute comparison for mainstream representative methods. Notations: Augmentation-based (Aug.), Embedding-based (Emb.), Reconstruction-based (Rec.), Parameters (Params), Memory (Mem.), Batch Size (BS), Optimizer (Optim.), Time (T.), ResNet (RN), Wide-ResNet (WRN), EfficientNet (EN), hours (h), minutes (m), seconds (s), unavailable (-), out-of-memory (OOM). Train and test time are evaluated under the standard setting described in Sec. 4.1 in one L40S GPU. Memory is tested under the standard setting with a batch size of 8, and the results for different methods are presented in Sec. 4.2. **Bold**, underline, and wavy-line represent the best, second-best, and third-best results, respectively.

| | Method | Hyper Params. | | | Efficiency | | | Train Mem. (M) | Test Mem. (M) | MVTec AD | | VisA | | Real-IAD | |
|---|---|---|---|---|---|---|---|---|---|---|---|---|---|---|---|
| | | BS | Optim. | LR | Params. | FLOPs | Backbone | | | Train T. | Test T. | Train T. | Test T. | Train T. | Test T. |
| Aug. | DRAEM Zavrtanik et al. (2021) | 8 | Adam | 1e-4 | 97.4M | 198G | UNet | 12,602 | 2,858 | 3.4h | 35s | 8.0h | 36s | 39.2 | 18m41s |
| | SimpleNet Liu et al. (2023) | 32 | AdamW | 1e-4 | 72.8M | 17.715G | WRN50 | 2,266 | 4,946 | 1.7h | 5m50s | 3.9h | 7m21s | 15.9h | 4h51m |
| | RealNet Zhang et al. (2024c) | 16 | Adam | 1e-4 | 591M | 115G | WRN50 | 14,004 | 3,794 | 2.6h | 41s | 5.4h | 41s | - | - |
| Emb. | CFA Lee et al. (2022) | 4 | AdamW | 1e-3 | 38.6M | 55.3G | WRN50 | 4,364 | 2,826 | 1.2h | 18s | 2.7h | 17s | 10.9h | 14m20s |
| | PatchCore Roth et al. (2022) | 8 | - | - | - | - | WRN50 | - | - | - | 9h22m | - | OOM | - | OOM |
| | CFLOW-AD Gudovskiy et al. (2022) | 32 | Adam | 2e-4 | 237M | 28.7G | WRN50 | 3,048 | 1,892 | 5.2h | 56s | 10.4h | 1m15s | 40.9h | 22m49s |
| | PyramidalFlow Lei et al. (2023) | 2 | Adam | 2e-4 | 34.3M | 962G | RN18 | 3,904 | 2,836 | 7.9h | 1m30s | 10.5h | 2m43s | 45h | 38m15s |
| Rec. | RD Deng & Li (2022) | 16 | Adam | 5e-3 | 80.6M | 28.4G | WRN50 | 3,286 | 1,464 | **0.8h** | **13s** | 1.1h | 18s | 4.1h | 7m48s |
| | DiAD He et al. (2024b) | 12 | Adam | 1e-5 | 1331M | 451.5G | RN50 | 26,146 | 20,306 | 9.1h | 16m | 21.6h | 19m | 90h | 16h20m |
| | ViTAD Zhang et al. (2023a) | 8 | AdamW | 1e-4 | 39.0M | 9.7G | ViT-S | **1,470** | **800** | **0.8h** | 15s | 1.1h | **15s** | 5.2h | 10m2s |
| | InvAD Zhang et al. (2024a) | 32 | Adam | 1e-3 | 95.6M | 45.4G | WRN50 | 5,920 | 3,398 | 1.0h | 31s | 2.3h | 33s | 9.2h | 21m |
| | InvAD-lite Zhang et al. (2024a) | 32 | Adam | 1e-3 | 17.1M | 9.3G | RN34 | 1,846 | 1,100 | 0.8h | 20s | 1.1h | 31s | **3.4h** | 9m27s |
| | MambaAD He et al. (2024a) | 16 | AdamW | 5e-3 | 25.7M | 8.3G | RN34 | 6,542 | 1,484 | 2.4h | 34s | 5.6h | 23s | 23.6h | 24m6s |
| Hybrid | UniAD You et al. (2022) | 8 | AdamW | 1e-4 | 24.5M | **3.4G** | EN-b4 | 1,856 | 968 | 0.8h | 22s | 1.0h | 18s | 4.1h | 7m2s |
| | RD++ Tien et al. (2023) | 16 | Adam | 1e-3 | 96.1M | 37.5G | WRN50 | 4,772 | 1,480 | 3.3h | 28s | 7.8h | 33s | 42.4h | 15m17s |
| | DesTSeg Zhang et al. (2023c) | 32 | SGD | 0.4 | 35.2M | 30.7G | RN18 | 3,446 | 1,240 | 1.0h | 19s | 1.7h | 16s | 6.8h | 8m13s |

et al. (2022); You et al. (2022); Zhang et al. (2023a) have attempted to incorporate this architecture into the design of anomaly detection models.

# 3 METHODOLOGY: ADER BENCHMARK

## 3.1 SUPPORTED VAD METHODS

Following the categories of current VAD methods in Sec. 2.3, we choose representative models for each category. The selection criteria are based on the method's popularity, effectiveness, and ease of use. *1) For Augmentation-based methods*, we choose DRAEM Zavrtanik et al. (2021), SimpleNet Liu et al. (2023), and RealNet Zhang et al. (2024c). *2) For Embedding-based methods*, we select CFA Lee et al. (2022), PatchCore Roth et al. (2022), CFLOW-AD Gudovskiy et al. (2022), and PyramidalFlow Lei et al. (2023). *3) For Reconstruction-based methods*, we include RD Deng & Li (2022), DiAD He et al. (2024b), ViTAD Zhang et al. (2023a), InvAD Zhang et al. (2024a), InvAD-lite Zhang et al. (2024a), and MambaAD He et al. (2024a). Additionally, UniAD You et al. (2022), RD++ Tien et al. (2023), and DesTSeg Zhang et al. (2023c) are categorized as *hybrid methods* due to their use of multiple techniques. Fig. 2 presents schematic diagrams and comparisons of the frameworks for each method belonging to different types, facilitating a better understanding of the differences among these methods. Tab. 1 provides a direct comparison of the hyperparameters, efficiency, and training time on three mainstream datasets for different methods, using one L40S GPU. Note that different methods may yield varying results when tested on different hardware, but the overall relative trends remain largely unchanged.

## 3.2 VAD DATASETS

To comprehensively evaluate the effectiveness, stability, and generalization of different methods, we benchmark extensive and fair experiments on three types of datasets: *1)* Real and synthetic industrial anomaly detection (AD) datasets, *i.e.*, MVTec AD Bergmann et al. (2019), MVTec AD 3D Bergmann et al. (2022b), MVTec LOCO-AD Bergmann et al. (2022a), VisA Zou et al. (2022), BTAD Mishra et al. (2021), MPDD Jezek et al. (2021), MAD_Real Zhou et al. (2024), MAD_Sim Zhou et al. (2024), and Real-IAD Wang et al. (2024). *2)* The medical Uni-Medical Zhang et al. (2023a) dataset. *3)* The general-purpose COCO-AD Zhang et al. (2024a) dataset. Detailed descriptions of the datasets are provided in Tab. 2, including the categories and scales of the datasets. Note that COCO-AD is inherently a multi-class dataset with four splits, and the average is taken when evaluating the comprehensive results.

Table 2: Comparison of representative VAD datasets, *i.e.*, industrial, medical, and general-purpose fields, respectively. Large-scale Real-IAD and COCO-AD only employ the 100 epoch setting.

| Dataset | Category Number | | Image Quantity | | | Epoch Setting in ADer | |
|---|---|---|---|---|---|---|---|
| | Train | Test | Train | Test | | | |
| | | | Normal | Anomaly | Normal | | |
| MVTec AD Bergmann et al. (2019) | 15 | 15 | 3,629 | 1,258 | 467 | 100 | 300 |
| MVTec AD 3D Bergmann et al. (2022b) | 10 | 10 | 2,950 | 249 | 948 | 100 | 300 |
| MVTec LOCO-AD Bergmann et al. (2022a) | 5 | 5 | 1,772 | 993 | 575 | 100 | 300 |
| VisA Zou et al. (2022) | 12 | 12 | 8,659 | 962 | 1,200 | 100 | 300 |
| BTAD Mishra et al. (2021) | 3 | 3 | 1,799 | 580 | 451 | 100 | 300 |
| MPDD Jezek et al. (2021) | 6 | 6 | 888 | 282 | 176 | 100 | 300 |
| MAD_Real Zhou et al. (2024) | 10 | 10 | 490 | 221 | 50 | 100 | 300 |
| MAD_Sim Zhou et al. (2024) | 20 | 20 | 4,200 | 4,951 | 638 | 100 | 300 |
| Real-IAD Wang et al. (2024) | 30 | 30 | 36,465 | 51,329 | 63,256 | 100 | - |
| Uni-Medical Zhang et al. (2023a) | 3 | 3 | 13,339 | 4,499 | 2,514 | 100 | 300 |
| COCO-AD Zhang et al. (2024a) | 61 | 81 | 30,438 | 1,291 | 3,661 | 100 | - |
| | | | 65,133 | 2,785 | 2,167 | 100 | - |
| | | | 79,083 | 3,328 | 1,624 | 100 | - |
| | | | 77,580 | 3,253 | 1,699 | 100 | - |

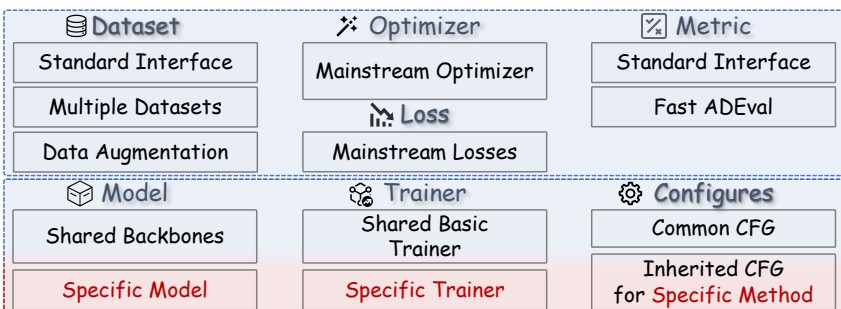

Figure 3: Core sub-modules of the training framework in ADer. The blue area represents standard components, while the red area indicates that a new method requires only three corresponding files.

## 3.3 EVALUATION METRICS

Following the ViTAD Zhang et al. (2023a) setting, we select image-level mean Area Under the Receiver Operating Curve (mAU-ROC) Zavrtanik et al. (2021), mean Average Precision (mAP) Zavrtanik et al. (2021), mean $F_1$-score (m$F_1$-max) Zou et al. (2022), region-level mean Area Under the Per-Region-Overlap (mAU-PRO) Bergmann et al. (2020), pixel-level mAU-ROC, mAP, m$F_1$-max, and the average AD (mAD) Zhang et al. (2023a) of seven metrics to evaluate all experiments. Additionally, we adopt the more practical order-independent pixel-level mean maximal Intersection over Union (mIoU-max) proposed in InvAD Zhang et al. (2024a).

## 3.4 SIMPLIFY IMPLEMENTATION BY STRUCTURED ADER CODEBASE

To ensure fair comparison among different methods, we construct a standardized ADer framework. As shown in Fig. 3, it includes shared foundational training/testing components and implements various metric calculations (compatible with our ADEval). The standardized dataset allows for easy comparison, eliminating potential unfair settings from different codebases. Additionally, ADer is highly extensible for new methods, requiring only compliant model, trainer, and configuration files.

## 3.5 FEATURE COMPARISON WITH CURRENT BENCHMARKS.

The existing vision anomaly detection benchmark works are primarily open-iad and anomalib. However, their updates for general AD models only extend up to 2022, and they **have not yet implemented the latest and practical multi-class anomaly detection methods**. We briefly discuss the relationship between the most popular Anomalib and ADer as follows: *1)* From the framework perspective: Anomalib is based on PyTorch Lightning that is deeply encapsulated, whereas ADer has a shallower encapsulation, exposing more interfaces to facilitate rapid algorithm iteration. *2)* From the methods perspective: Anomalib supports general AD models only up to 2022, while ADer supports

Table 3: Benchmarked results on MVTec AD dataset Bergmann et al. (2019) by the suggested metrics in Sec. 3.3 under 100/300 epochs. **Bold**, underline, and wavy-line represent the best, second-best, and third-best results, respectively. Patchcore requires no training that shares results under different epoch settings.

| | Method | Image-level | | | Pixel-level | | | mAU-PRO | mIoU-max | mAD |
|---|---|---|---|---|---|---|---|---|---|---|
| | | mAU-ROC | mAP | mF$_1$-max | mAU-ROC | mAP | mF$_1$-max | | | |
| Aug. | DRAEM Zavrtanik et al. (2021) | 54.5/55.2 | 76.3/77.0 | 83.6/83.9 | 47.6/48.7 | 3.2/3.1 | 6.7/6.3 | 14.3/15.8 | 3.5/3.3 | 44.7/45.3 |
| | SimpleNet Liu et al. (2023) | 95.4/79.2 | 98.3/90.8 | 95.7/87.6 | 96.8/82.4 | 48.8/24.0 | 51.9/29.0 | 86.9/62.0 | 36.4/17.8 | 83.8/67.6 |
| | RealNet Zhang et al. (2024c) | 84.8/82.9 | 94.1/93.3 | 90.9/90.9 | 72.6/69.8 | 48.2/50.0 | 41.4/40.4 | 56.8/51.2 | 28.8/28.5 | 72.3/70.9 |
| Emb. | CFA Lee et al. (2022) | 57.6/55.8 | 78.3/78.8 | 84.7/84.5 | 54.8/43.9 | 11.9/4.8 | 14.7/8.9 | 25.3/19.3 | 8.9/4.7 | 50.1/46.1 |
| | PatchCore Roth et al. (2022) | 98.8/- | 99.5/- | 98.4/- | 98.3/- | 59.9/- | 61.0/- | 94.2/- | 44.9/- | 88.6/- |
| | CFLOW-AD Gudovskiy et al. (2022) | 91.6/92.7 | 96.7/97.2 | 93.4/94.0 | 95.7/95.8 | 45.9/46.8 | 48.6/49.6 | 88.3/89.0 | 33.2/34.0 | 81.8/82.5 |
| | PyramidalFlow Lei et al. (2023) | 70.2/66.2 | 85.5/84.3 | 85.5/85.1 | 80.0/74.2 | 22.3/17.2 | 22.0/19.6 | 47.5/40.0 | 12.8/11.4 | 61.7/58.1 |
| Rec. | RD Deng & Li (2022) | 93.6/90.5 | 97.2/95.0 | 95.6/95.1 | 95.8/95.9 | 48.2/47.1 | 53.6/52.1 | 91.2/91.2 | 37.0/35.8 | 83.8/82.6 |
| | DiAD He et al. (2024b) | 88.9/92.0 | 95.8/96.8 | 93.5/94.4 | 89.3/89.3 | 27.0/27.3 | 32.5/32.7 | 63.9/64.4 | 21.1/21.3 | 70.1/71.0 |
| | ViTAD Zhang et al. (2023a) | 98.3/98.4 | 99.3/99.4 | 97.3/97.5 | 97.6/97.5 | 55.2/55.2 | 58.4/58.1 | 92.0/91.7 | 42.3/42.0 | 87.1/87.0 |
| | InvAD Zhang et al. (2024a) | 98.1/98.9 | 99.0/99.6 | 97.6/98.2 | 98.0/98.1 | 56.3/57.1 | 59.2/59.6 | 94.4/94.4 | 42.8/43.1 | 87.6/88.1 |
| | InvAD-lite Zhang et al. (2024a) | 97.9/98.1 | 99.2/99.1 | 96.8/96.8 | 97.3/97.3 | 54.4/55.0 | 57.8/58.1 | 93.3/93.1 | 41.4/41.7 | 86.8/86.9 |
| | MambaAD He et al. (2024a) | 97.8/98.5 | 99.3/99.5 | 97.3/97.7 | 97.4/97.6 | 55.1/56.1 | 57.6/58.7 | 93.4/93.6 | 41.2/42.3 | 87.0/87.5 |
| Hybrid | UniAD You et al. (2022) | 92.5/96.8 | 97.3/98.9 | 95.4/97.0 | 95.8/96.8 | 42.7/45.0 | 48.0/50.2 | 89.3/91.0 | 32.5/34.2 | 82.0/84.2 |
| | RD++ Tien et al. (2023) | 97.9/95.8 | 98.8/98.0 | 96.4/96.6 | 97.3/97.3 | 54.7/53.0 | 58.0/57.0 | 93.2/92.9 | 41.5/40.5 | 86.8/85.9 |
| | DesTSeg Zhang et al. (2023c) | 96.4/96.3 | 98.6/98.8 | 96.2/96.1 | 92.0/92.6 | 71.1/75.8 | 68.2/71.3 | 83.4/82.6 | 52.8/56.6 | 87.9/88.8 |

Table 4: Benchmarked results on VisA dataset Zou et al. (2022) by the suggested metrics under 100/300 epochs.

| | Method | Image-level | | | Pixel-level | | | mAU-PRO | mIoU-max | mAD |
|---|---|---|---|---|---|---|---|---|---|---|
| | | mAU-ROC | mAP | mF$_1$-max | mAU-ROC | mAP | mF$_1$-max | | | |
| Aug. | DRAEM Zavrtanik et al. (2021) | 55.1/56.2 | 62.4/64.6 | 72.9/74.9 | 37.5/45.0 | 0.6/0.7 | 1.7/1.8 | 10.0/16.0 | 0.9/0.9 | 38.0/40.6 |
| | SimpleNet Liu et al. (2023) | 86.4/80.7 | 89.1/83.8 | 82.8/79.3 | 96.6/94.4 | 34.0/29.2 | 37.8/33.1 | 79.2/74.2 | 25.7/22.1 | 74.0/69.5 |
| | RealNet Zhang et al. (2024c) | 71.4/79.2 | 79.5/84.8 | 74.7/78.3 | 61.0/65.4 | 25.7/29.2 | 22.6/27.9 | 27.4/33.9 | 13.5/17.4 | 54.7/59.9 |
| Emb. | CFA Lee et al. (2022) | 66.3/67.1 | 74.3/73.8 | 74.2/75.3 | 81.3/83.0 | 22.1/13.7 | 26.2/18.7 | 50.8/48.7 | 17.0/11.3 | 58.4/56.5 |
| | CFLOW-AD Gudovskiy et al. (2022) | 86.5/87.2 | 88.8/89.5 | 84.9/85.1 | 97.7/97.8 | 33.9/34.2 | 37.2/37.2 | 86.8/87.3 | 24.9/24.9 | 75.3/75.7 |
| | PyramidalFlow Lei et al. (2023) | 58.2/69.0 | 66.3/72.9 | 74.4/75.8 | 77.0/79.1 | 7.2/7.9 | 9.6/8.7 | 42.8/52.6 | 5.6/4.7 | 50.2/54.8 |
| Rec. | RD Deng & Li (2022) | 90.6/93.9 | 90.9/94.8 | 89.3/90.4 | 98.0/98.1 | 35.4/38.4 | 42.5/43.7 | 91.9/91.9 | 27.9/29.0 | 78.6/80.5 |
| | DiAD He et al. (2024b) | 84.8/90.5 | 88.5/91.4 | 86.9/90.4 | 82.5/83.4 | 17.9/19.2 | 23.2/25.0 | 44.5/44.3 | 14.9/16.2 | 61.2/63.5 |
| | ViTAD Zhang et al. (2023a) | 90.4/90.3 | 91.1/91.2 | 86.0/86.4 | 98.2/98.2 | 36.4/36.4 | 41.0/40.9 | 85.7/85.8 | 27.5/27.5 | 77.2/77.3 |
| | InvAD Zhang et al. (2024a) | 95.4/95.6 | 95.7/96.0 | 91.6/92.3 | 98.9/99.0 | 43.3/43.7 | 46.8/46.9 | 93.1/93.0 | 32.5/32.6 | 82.4/82.6 |
| | InvAD-lite Zhang et al. (2024a) | 94.9/95.3 | 95.2/95.8 | 90.7/91.0 | 98.6/98.7 | 40.2/41.2 | 44.0/44.9 | 93.1/93.2 | 29.8/30.6 | 81.3/81.8 |
| | MambaAD He et al. (2024a) | 94.5/93.6 | 94.9/93.9 | 90.2/89.8 | 98.4/98.2 | 39.3/34.0 | 43.7/39.3 | 92.1/90.5 | 29.5/25.9 | 80.8/79.0 |
| Hybrid | UniAD You et al. (2022) | 89.0/91.4 | 91.0/93.3 | 85.8/87.5 | 98.3/98.5 | 34.5/35.3 | 39.6/40.2 | 86.5/89.0 | 26.4/26.5 | 76.7/78.2 |
| | RD++ Tien et al. (2023) | 93.9/93.1 | 94.7/94.1 | 90.2/90.0 | 98.4/98.4 | 42.3/40.4 | 46.3/44.8 | 91.9/91.4 | 31.2/29.9 | 81.3/80.6 |
| | DesTSeg Zhang et al. (2023c) | 89.9/89.0 | 91.4/90.3 | 86.7/85.9 | 86.7/84.8 | 46.6/43.3 | 47.2/44.4 | 61.1/57.5 | 32.7/30.1 | 74.9/73.0 |

more recent models up to 2024. *3)* From the data and metrics perspective: Compared to Anomalib, ADer supports large-scale industrial Real-IAD Wang et al. (2024), medical Uni-Medical Zhang et al. (2023a), and general-purpose COCO-AD Zhang et al. (2024a) datasets, as well as more application-relevant metrics like mIoU-max Zhang et al. (2024a) and averaged mAD Zhang et al. (2023a). *4)* From the setting perspective: ADer focuses more on the recently popular and future research trend of multi-class settings.

## 3.6 ADEVAL: FAST AND MEMORY-EFFICIENT ROUTINES FOR MAU-ROC/MAP/MAU-PRO

The speed of metric evaluation is crucial for the iterative process of model algorithms. When the number of test images increases and the resolution becomes higher, the pixel-level evaluation algorithms implemented naively using *sklearn* and *skimage* packages become time-consuming. This is particularly evident with large-scale datasets such as Real-IAD Wang et al. (2024) and COCO-AD Zhang et al. (2024a), where evaluation times can exceed one hour. To address this issue, we have released the GPU-assisted ***ADEval*** library, which employs an iterative-accumulating algorithm with optional CUDA acceleration. For instance, in the case of the most time-consuming mAU-PRO metric on the multi-class MVTec AD dataset, as demonstrated by UniAD You et al. (2022), the naive implementation requires *242.7 seconds*, whereas the optimized version reduces this to less than *0.1 second*, achieving a more than 1000-fold speedup.

Table 5: Benchmarked results on Real-IAD dataset Wang et al. (2024) by the suggested metrics under 100 epoch.

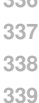

| Method | Image-level | | | Pixel-level | | | mAU-PRO | mIoU-max | mAD |
|---|---|---|---|---|---|---|---|---|---|
| | mAU-ROC | mAP | mF$_1$-max | mAU-ROC | mAP | mF$_1$-max | | | |
| **Aug.** DRAEM Zavrtanik et al. (2021) | 50.9 | 45.9 | 61.3 | 44.0 | 0.2 | 0.4 | 13.6 | 0.2 | 33.6 |
| SimpleNet Liu et al. (2023) | 54.9 | 50.6 | 61.5 | 76.1 | 1.9 | 4.9 | 42.4 | 2.5 | 43.5 |
| **Emb.** CFA Lee et al. (2022) | 55.7 | 50.5 | 61.9 | 81.3 | 1.6 | 3.8 | 48.8 | 2.0 | 45.0 |
| CFLOW-AD Gudovskiy et al. (2022) | 77.0 | 75.8 | 69.9 | 94.8 | 17.6 | 21.7 | 80.4 | 12.4 | 63.9 |
| PyramidalFlow Lei et al. (2023) | 54.4 | 48.0 | 62.0 | 71.1 | 1.2 | 1.1 | 34.9 | 0.5 | 40.9 |
| **Rec.** RD Deng & Li (2022) | 82.7 | 79.3 | 74.1 | 97.2 | 25.2 | 32.8 | 90.0 | 20.0 | 70.0 |
| DiAD He et al. (2024b) | 75.6 | 66.4 | 69.9 | 88.0 | 2.9 | 7.1 | 58.1 | 3.7 | 52.6 |
| ViTAD Zhang et al. (2023a) | 82.7 | 80.2 | 73.7 | 97.2 | 24.3 | 32.3 | 84.8 | 19.6 | 69.3 |
| InvAD Zhang et al. (2024a) | **89.4** | **87.0** | **80.2** | 98.4 | 32.6 | 38.9 | **92.7** | 24.6 | **75.6** |
| InvAD-lite Zhang et al. (2024a) | 87.2 | 85.2 | 77.8 | 98.0 | 31.7 | 37.9 | 92.0 | 23.8 | 74.2 |
| MambaAD He et al. (2024a) | 87.0 | 85.3 | 77.6 | **98.6** | 32.4 | 38.1 | 91.2 | 23.9 | 74.2 |
| **Hybrid** UniAD You et al. (2022) | 83.1 | 81.2 | 74.5 | 97.4 | 23.3 | 30.9 | 87.1 | 18.6 | 69.6 |
| RD++ Tien et al. (2023) | 83.6 | 80.6 | 74.8 | 97.7 | 25.9 | 33.6 | 90.7 | 20.5 | 70.8 |
| DesTSeg Zhang et al. (2023c) | 79.3 | 76.7 | 70.7 | 80.3 | **36.9** | **40.3** | 56.1 | **26.2** | 64.5 |

Table 6: Benchmarked results on all other datasets by the mIoU/mAD metrics under 100 epoch.

| Method | MVTec 3D | MVTec LOCO | BTAD | MPDD | MAD_Real | MAD_Sim | Uni-Medical | COCO-AD |
|---|---|---|---|---|---|---|---|---|
| **Aug.** DRAEM Zavrtanik et al. (2021) | 1.0/46.6 | 5.6/41.8 | 3.4/47.4 | 2.5/35.4 | 0.8/48.9 | 0.7/48.0 | 3.0/37.0 | 8.0/38.3 |
| SimpleNet Liu et al. (2023) | 13.9/70.1 | 21.2/67.4 | 28.6/78.8 | 24.5/76.2 | 6.3/56.7 | 4.2/60.9 | 23.3/68.9 | 11.5/42.9 |
| RealNet Zhang et al. (2024c) | - / - | - / - | 36.6/76.1 | 28.2/72.3 | - / - | - / - | - / - | - / - |
| **Emb.** CFA Lee et al. (2022) | 9.3/58.6 | 9.1/52.2 | 33.6/80.2 | 16.6/65.1 | 8.7/58.7 | 4.6/58.2 | 14.7/57.2 | 8.9/41.0 |
| PatchCore Roth et al. (2022) | 24.5/78.4 | 20.4/67.7 | 38.0/83.3 | 35.0/83.5 | 16.6/68.2 | - / - | - / - | - / - |
| CFLOW-AD Gudovskiy et al. (2022) | 15.8/71.6 | 17.3/64.0 | 33.8/78.9 | 20.1/69.7 | 8.8/63.4 | 2.7/60.0 | 17.7/69.9 | 16.0/53.0 |
| PyramidalFlow Lei et al. (2023) | 6.4/61.9 | 8.0/47.5 | 18.3/68.3 | 10.4/64.6 | 5.1/59.0 | 2.5/57.4 | 9.4/47.6 | 8.0/38.4 |
| **Rec.** RD Deng & Li (2022) | 22.2/75.3 | 15.8/63.0 | 42.1/84.7 | 31.4/80.6 | 7.2/60.6 | 4.5/62.6 | 26.9/71.1 | 11.5/45.8 |
| DiAD He et al. (2024b) | 5.4/62.8 | 14.9/56.5 | 15.7/68.5 | 8.2/58.1 | 3.6/55.8 | 4.2/58.9 | 23.2/69.1 | 11.6/44.1 |
| ViTAD Zhang et al. (2023a) | 20.4/75.4 | 19.8/64.8 | 40.1/83.1 | 27.7/77.3 | 5.0/57.4 | 5.0/62.8 | 33.7/75.5 | 20.1/53.5 |
| InvAD Zhang et al. (2024a) | 27.4/80.3 | 23.1/69.2 | **44.3/85.9** | 34.0/81.9 | 16.8/67.6 | 8.6/67.4 | 32.6/75.4 | 14.3/50.5 |
| InvAD-lite Zhang et al. (2024a) | 26.9/79.6 | 20.6/67.1 | 42.6/84.4 | 30.9/79.8 | 8.1/62.5 | 6.0/64.9 | 26.5/71.6 | 13.7/49.2 |
| MambaAD He et al. (2024a) | 25.9/79.0 | 20.6/66.4 | 39.0/82.5 | 26.8/78.1 | 7.2/60.5 | 5.0/63.6 | 33.5/75.9 | 12.9/48.8 |
| **Hybrid** UniAD You et al. (2022) | 16.7/72.3 | 21.6/64.4 | 36.9/82.9 | 12.5/63.3 | 5.8/58.2 | 3.5/60.9 | 27.6/71.6 | 10.9/44.0 |
| RD++ Tien et al. (2023) | 25.2/78.1 | 17.6/64.5 | 42.8/84.8 | 33.6/80.8 | 8.5/61.4 | 4.4/62.7 | 29.4/72.3 | 11.8/46.4 |
| DesTSeg Zhang et al. (2023c) | **28.4**/72.6 | 20.3/63.9 | 29.0/77.0 | 25.6/71.9 | 4.5/52.8 | 4.1/53.6 | 21.2/61.0 | 8.5/40.7 |

## 4 RESULTS AND ANALYSIS

### 4.1 EXPERIMENTAL SETUP

Different methods potentially introduce various factors that can impact model performance. To ensure a fair and comprehensive evaluation of the effectiveness and convergence of different methods, we fix the most influential parameters, *i.e.*, resolution (256×256) and training epochs (100 and 300). The reason lies in the fact that tasks such as classification, detection, and segmentation typically set specific resolutions and standard training epochs. We observe that for most methods, 100 epochs generally suffice to reach saturation Zhang et al. (2023a; 2024a), with only a few methods You et al. (2022) requiring more epochs for training. Therefore, we also establish a setting with 300 epochs. Meanwhile, we maintain consistency with the original papers for batch size, optimizer, learning rate, and data augmentation. We report the evaluation results corresponding to the final epoch at the end of training to ensure fairness. All experiments are conducted on one L40S GPU.

### 4.2 BENCHMARK RESULTS ON INDUSTRIAL, MEDICAL, AND GENERAL-PURPOSE UAD DATASETS

To thoroughly evaluate the effectiveness of different methods and their adaptability to various data domains, we conduct experiments on multiple datasets across three domains. Due to space constraints, we report the average metrics for the popular MVTec AD (see Tab. 3), VisA (see Tab. 4), and Real-IAD (see Tab. 5) datasets in the main paper. For the remaining datasets, we report the mAD and mIoU-max metrics (see Tab. 6). Full results for each category are provided in Appendix **??** to **??**.

**Quantitative results.** InvAD Zhang et al. (2024a) consistently shows excellent performance across all datasets. ViTAD Zhang et al. (2023a) and MambaAD He et al. (2024a), specifically designed for multi-class settings, also achieve good results. In contrast, DiAD He et al. (2024b) and UniAD You

et al. (2022) require more epochs to converge and do not perform well under the 100/300 epoch standard we set. DeSTSeg Zhang et al. (2023c) exhibits outstanding performance in pixel-level segmentation. Methods designed for single-class settings, such as RD Deng & Li (2022), RD++ Tien et al. (2023), CFLOW-AD Gudovskiy et al. (2022), and RealNet Zhang et al. (2024c), also perform well in multi-class settings. However, single-class methods like DRAEM Zavrtanik et al. (2021), SimpleNet Liu et al. (2023), CFA Lee et al. (2022), and PyramidFlow Lei et al. (2023) show significant performance gaps in multi-class anomaly detection and are not suitable for such tasks. Considering the training time, model parameters, and FLOPs shown in Fig. 2, InvAD, InvAD-lite, and ViTAD achieve a good balance of effectiveness and efficiency. RD, UniAD, and DeSTSeg also perform well in terms of both efficiency and effectiveness. On the other hand, methods like DiAD, PyramidFlow, CFLOW-AD, RD++, RealNet, MambaAD, and SimpleNet have significantly longer training times compared to other methods.

**Qualitative results.** Fig. 4 presents intuitive visualization results under the training setting of 100 epochs on popular MVTec AD Bergmann et al. (2019) and VisA Zou et al. (2022) datasets, as well as the medical Uni-Medical Zhang et al. (2023a) and large-scale Real-IAD Wang et al. (2024) datasets.

**Convergence analysis.** From Tab. 3 and Tab. 4, we analyze the convergence of different methods by comparing the results after training for 100 epochs and 300 epochs. The methods can be categorized into three groups: 1) Methods that show no significant improvement in performance after 300 epochs compared to 100 epochs, indicating rapid convergence within 100 epochs. These models include DRAEM Zavrtanik et al. (2021), SimpleNet Liu et al. (2023), CFA Lee et al. (2022), PyramidFlow Lei et al. (2023), ViTAD Zhang et al. (2023a), InvAD-lite Zhang et al. (2024a), RD++ Tien et al. (2023), and DeSTSeg Zhang et al. (2023c).2) Methods that show improvement with continued training on the VisA dataset but no improvement or a decline on the MVTec AD dataset, indicating slower convergence on larger datasets. These models include RealNet and RD. 3) Methods that show significant improvement after 300 epochs compared to 100 epochs, indicating slower convergence. These models include CFLOW-AD Gudovskiy et al. (2022), DiAD He et al. (2024b), InvAD Zhang et al. (2024a), MambaAD He et al. (2024a), and UniAD You et al. (2022).

**Stability analysis.** For current anomaly detection algorithms, most authors select the best epoch's results as the model's performance. However, this method of epoch selection is unscientific and may indicate significant model instability. Therefore, we further analyze model stability using Tab. 3 and Tab. 4, comparing the results at 100 epochs and 300 epochs to identify any substantial differences. The results show that SimpleNet Liu et al. (2023) and PyramidFlow Lei et al. (2023) exhibit considerable differences, indicating poor model stability, while other methods do not show significant fluctuations.

**Cross-domain dataset correlation.** To analyze the adaptability of different methods across various datasets and the relationships and differences between different types of datasets, we employ Pearson correlation analysis to examine the correlations among these datasets. Specifically, we select four distinct datasets for analysis: MVTec AD Bergmann et al. (2019), Uni-Medical Zhang et al. (2023a), Real-IAD Wang et al. (2024), and COCO-AD Zhang et al. (2024a). MVTec AD represents a fundamental industrial dataset, Uni-Medical consists of medical images from CT scans, Real-IAD is a large-scale multi-view industrial dataset from real-world scenarios, and COCO-AD is a large-scale panoptic segmentation dataset from real-life scenes. We evaluate four categories of methods using eight metrics: image and pixel mAU-ROC,

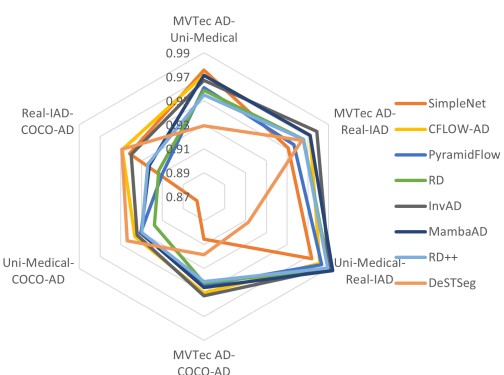

Figure 5: A Pearson correlation coefficient analysis for different methods on several datasets.

mAP, mF1-max, region mAU-PRO, and segmentation mIoU-max. The results, as shown in Fig. 5, indicate that the COCO-AD and Uni-Medical datasets exhibit lower Pearson correlation coefficients due to significant differences in data distribution compared to general industrial datasets. Although the Pearson correlation coefficient between the Uni-Medical and Real-IAD datasets is relatively high, Tab. 5 and Tab. ?? in the Appendix reveal that this is because all methods perform poorly on these two datasets. Additionally, it is observed that the methods SimpleNet and DeSTSeg show considerable

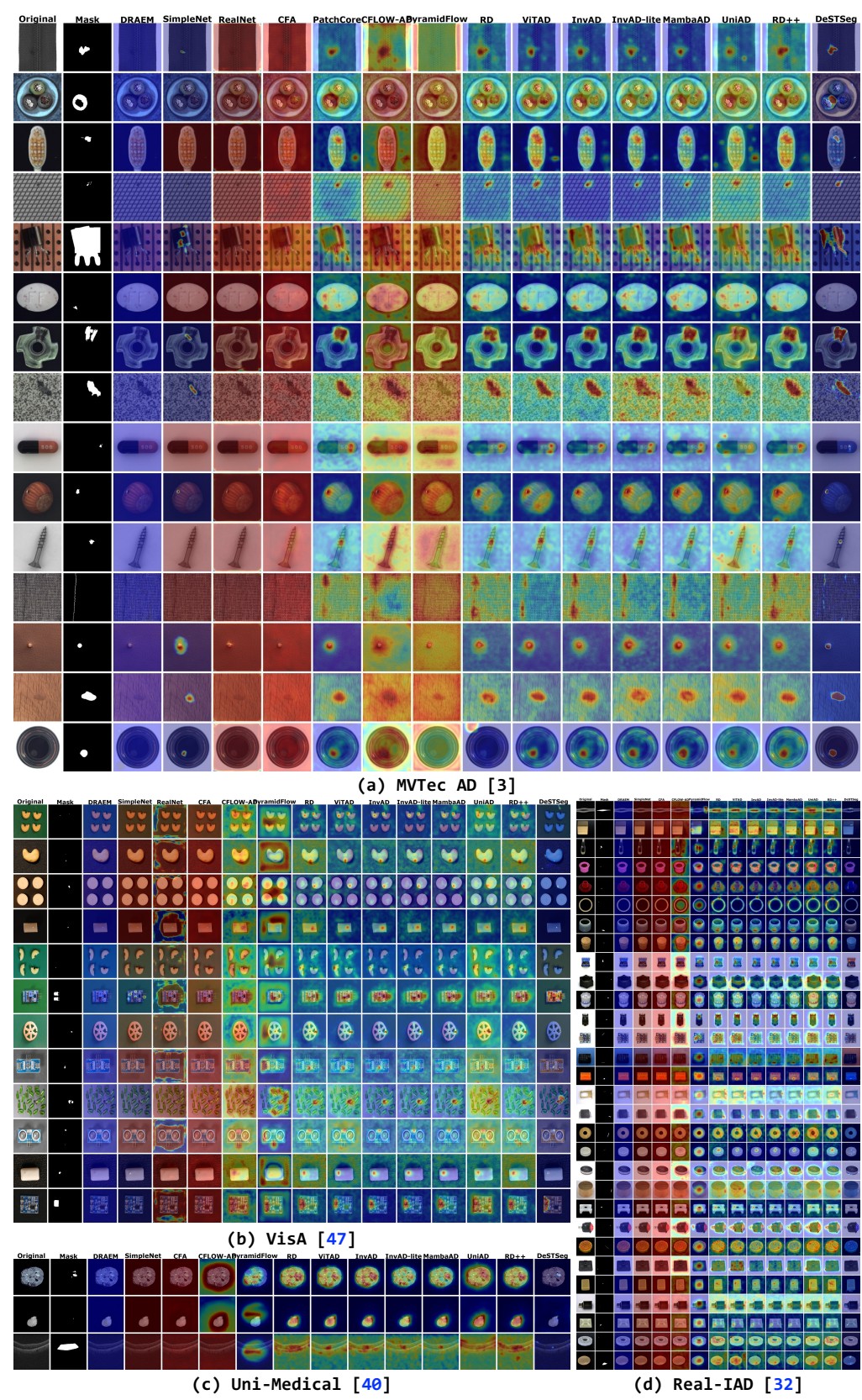

Figure 4: Qualitative visualizations on the popular MVTec AD and VisA datasets, as well as the medical Uni-Medical and large-scale Real-IAD datasets. Zoom in for better viewing.

instability in their results across different datasets. This instability may be attributed to the inherent instability of the data augmentation algorithms they employ.

**Training-free PatchCore.** PatchCore Roth et al. (2022) does not require model training. It extracts all features from the training data, then selects a core subset and stores it in a Memory Bank. During testing, each test image is compared with the Memory Bank to compute an anomaly score. Because it stores the core subset of all normal features in the Memory Bank, PatchCore is only feasible for multi-class anomaly detection tasks on small-scale datasets. For large-scale datasets, it faces limitations due to insufficient GPU and memory resources. Although it achieves excellent results on the MVTec AD dataset, as shown in Tab. 1, its testing time is nearly a thousand times longer than other methods. In summary, PatchCore performs exceptionally well on small-scale datasets but is constrained by large-scale datasets and testing time.

**Dataset Analysis.** The experimental results indicate that there is room for improvement in the VisA Zou et al. (2022) and Real-IAD Wang et al. (2024) datasets due to the very small defect areas, necessitating models with stronger capabilities for detecting minor defects. The MAD_Real and MAD_Sim Zhou et al. (2024) datasets, due to their small data volume and varying difficulty levels, result in similar performance across all models, particularly in the mF1-max metric. The Uni-Medical Zhang et al. (2023a) dataset, consisting of images converted from CT scans, has a data distribution that significantly differs from other industrial datasets, suggesting the need for specialized detection networks tailored to medical datasets. COCO-AD Zhang et al. (2024a), as a newly proposed large-scale dataset for general scenarios, presents high complexity. Current industrial AD networks are unable to achieve effective results on the COCO-AD dataset.

### 4.3 Challenges for Current VAD

**Immature method.** For challenging anomaly detection datasets such as MVTecLOCO, pose-agnostic MAD, and general-purpose COCO-AD, current methods perform poorly in a multi-class setting. Future research should focus on designing more robust methods to address this issue.

**Efficiency.** Most methods do not consider model complexity during design, resulting in high FLOPs. This issue becomes more pronounced when applied to real-world high-resolution scenarios. Incorporating lightweight characteristics in model design could be a potential solution.

**Dataset scale.** Mainstream datasets in the VAD field, such as MVTec AD and Real-IAD, are relatively small compared to those in detection and segmentation fields and are tailored to specific industrial scenarios. This limitation could hinder technological development. Collecting larger-scale, general-scene AD datasets is crucial for the advancement of the VAD field.

**VAD-specific metric.** Metrics like mAU-ROC and mAP are not uniquely designed for the BAD field. Developing more reliable evaluation methods to better meet practical application needs is essential.

**Augmentation and tricks.** In fields such as classification, detection, and segmentation, data augmentation and tricks are extremely important for model training. However, few studies explore their role in the AD field, potentially limiting model performance.

**Model interpretability.** In many applications, understanding why a model detects a particular anomaly is crucial. Providing effective visualization tools to display detection results and the model's decision-making process remains a challenge.

## 5 Conclusion and Discussion

This paper addresses the urgent need for a comprehensive and fair benchmark in the field of visual anomaly detection. We introduce a modular and scalable *ADer* library designed to fairly facilitate the evaluation of fifteen advanced VAD methods across multiple mainstream datasets. ensuring a thorough and unbiased assessment of each method's performance. Our extensive experiments reveal the strengths and weaknesses of different methods, providing valuable insights into their efficiency and training resource consumption. We also develop and open-source a GPU-assisted *ADEval* package to reduce the evaluation time, enabling extensive assessments. Experimental results highlight the challenges of various VAD methods and offer valuable insights for future research directions.

**Broader Impacts.** The open-sourcing ADer can accelerate the development of new VAD technology for the open-source community and become a valuable resource for practitioners in the field.

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
