# OpenReview forum: "A Comprehensive Library for Benchmarking Multi-class Visual Anomaly Detection"
_ICLR.cc/2025/Conference — ICLR 2025 Conference Withdrawn Submission_

### Official Review · Reviewer_5Nhj · 2024-10-29

**Soundness:** 2
**Presentation:** 1
**Contribution:** 2
**Rating:** 3
**Confidence:** 4

**Summary:**

Anomaly detection has been a popular topic in image processing for several years, with applications in a variety of fields (industrial and medical are mentioned here). The article notes that many methods have been proposed to address the problem in a specialized or general way, and that with the wide variety of applications, databases, method types, and metrics, it is difficult to assess the strengths, weaknesses, and relevance of these methods in a general way.

The authors of this article have developed a library, ADer, to provide a unified evaluation of recent anomaly detection methods. The result of this evaluation is presented in this article and is intended to help identify the strengths and weaknesses of anomaly detection methods through an evaluation on a variety of datasets and metrics.

The authors identified four main categories of methods and selected 15 for their evaluation, with at least 3 representatives per category. They tested them on 11 industrial, medical, and general databases. The benchmarking was done with 9 different metrics. The authors say that they worked on the practical implementation of these metrics to make their computation more efficient.

The experimental part presents the results of this benchmark in the form of several tables. The evaluation analyzes the quality of detection, but also the efficiency of the training and inference methods, as well as their stability.

**Strengths:**

The paper proposes a library of tools for unified and efficient evaluation of anomaly detection methods. As the authors note, the resources available for evaluation are not efficiently implemented. This can lead to two problems, either a waste of time due to inefficient evaluation, or a non-reproducible and inconsistent evaluation due to differences in metric implementation. The provision of a unified and efficient tool in this article will address both of these problems if the tool is adopted by the community.

Using this library that they have developed, the authors provide us with a series of tables that allow us to compare the different qualities of anomaly detection models. The tables are easy to read and well presented. The metrics allow us to evaluate the quality of detection on several levels, but also the time/memory efficiency of training and inference. This abundance of metrics allows for a unified analysis of the methods (which, however, is mainly accessible to experts in the field).

In addition to the quality and effectiveness metrics, I appreciated that the authors emphasized the stability of the methods by mentioning that the articles tended to present the best results obtained over a number of trials. It is important for a method to be stable and provide consistent results between different training sessions.

**Weaknesses:**

The article compares many methods, on many datasets, with many metrics to give a complete assessment. However, the description of these methods, datasets, and metrics is very superficial. The authors should provide a brief overview of the key innovations for each method, summarize the distinguishing characteristics of the datasets, and explain the strengths and limitations of each evaluation metric.

The methods are introduced in one paragraph and then quickly grouped into four broad categories. The article lacks a more detailed description of the methods that are later evaluated. We don't know the order in which these methods were introduced, or what deficiencies they attempted to correct with new design proposals. This information is important for understanding the evaluation tables and for validating or invalidating the claims of the methods. The lack of this information makes the analysis much more difficult for someone without a detailed and up-to-date knowledge of the field. However, this type of benchmark article is also usually intended to serve as a starting point for researchers and engineers entering the field or doing implementations.

The same comment applies to datasets, which are also introduced in a paragraph and a table. Aside from the "Dataset Analysis" paragraph at the end of the article, there is little information about what makes datasets different. We don't know how they work, how diverse they are, or what they actually represent. It would be desirable to have a qualitative comparison of the datasets from the beginning to better understand the evaluation. As mentioned above, Mvtec-AD is simple and can be considered a solved dataset, while Visa and RealIAD are more difficult. As the authors have chosen to present the tables of these datasets in the article, the differences should be presented from the beginning.

Finally, since this paper points out that the methods used are not very informative and suggests the use of others, they should be explained. There are 9 metrics used in this paper, and it would be necessary to explain the advantages of one metric over another. We need to know if they are complementary and what their variance and invariance are, otherwise the tables can't be analyzed in detail.

In the paragraph "Training-free PatchCore" it says: "PatchCore Roth et al. (2022) does not require model training. It extracts all features from the training data, then selects a core subset and stores it in a Memory Bank. During testing, each test image is compared with the Memory Bank to compute an "anomaly score"", which is not exactly the case. Patchcore has a training procedure, which is the extraction and sub-sampling of features from the train base. This procedure is performed only once, and then the resulting memory bank is simply accessed to perform a nearest-neighbor measurement with new features. In fact, there's a problem in Table 1, because Patchcore should have a training time. The authors should revise their description of PatchCore to accurately reflect its feature extraction and sub-sampling process, and they should include the training time for PatchCore in Table 1.

Finally, the authors claim that Patchcore can't run on datasets larger than Mvtec-AD, but this doesn't seem to be the case, as articles [1,2] with Patchcore's AUROC for VISA can be found. It's important to evaluate Patchcore on all datasets, as it's one of the best and one of the only ones that doesn't involve training a network. The authors should either evaluate PatchCore on all datasets or provide a clear explanation for why it was not evaluated on certain datasets, citing any technical limitations or resource constraints they encountered.

[1] Fučka, Matic, Vitjan Zavrtanik, and Danijel Skočaj. "TransFusion–A transparency-based diffusion model for anomaly detection." European Conference on Computer Vision. Springer, Cham, 2025.

[2] Zhang, Hui, et al. "DiffusionAD: Norm-guided one-step denoising diffusion for anomaly detection." arXiv preprint arXiv:2303.08730 (2023).

For reconstruction methods, with the current trend in image generation by diffusion, methods using it have been proposed. One is shown in this article, but others [2] that announce state-of-the-art performance should be included.  In general, some methods [2,3,4] that have a high AUROC on VISA should be included, since VISA seems to be the next dataset to be solved after Mvtec for the industrial problem.  The authors should consider including these specific diffusion-based methods and other high-performing methods, or they should explain their criteria for method selection if these were intentionally excluded.

[3] Fučka, Matic, Vitjan Zavrtanik, and Danijel Skočaj. "TransFusion–A transparency-based diffusion model for anomaly detection." European Conference on Computer Vision. Springer, Cham, 2025.

[4] Batzner, Kilian, Lars Heckler, and Rebecca König. "Efficientad: Accurate visual anomaly detection at millisecond-level latencies." Proceedings of the IEEE/CVF Winter Conference on Applications of Computer Vision. 2024.

**Questions:**

As a general comment, I'd recommend that authors better describe the methods, datasets and metrics they use, even if it means using less of them. This is important so that a non-expert reader can properly analyze the numerous tables in the article and understand the conclusions.

---

### Official Review · Reviewer_vKGS · 2024-11-02

**Soundness:** 2
**Presentation:** 2
**Contribution:** 3
**Rating:** 3
**Confidence:** 4

**Summary:**

This paper presents ADer, a new modular and extensible benchmark in Visual Anomaly Detection (VAD) field encompassing 15 state-of-the-art methods and 9 evaluation metrics across industrial, medical, and general-purpose datasets. Comprehensive experiments highlight the strengths and limitations of various methods, providing valuable insights into existing challenges and future research directions. The authors intend for ADer to serve as a crucial resource for advancing robust and generalizable visual anomaly detection systems.

**Strengths:**

1. ADer, a comprehensive and fair benchmark, is proposed for the Visual Anomaly Detection (VAD) field to foster its sustainable and healthy development.
2. ADer offers a convenient and fair approach for evaluating anomaly detection methods. It is designed as a highly scalable, modular framework that seamlessly integrates with existing techniques. The framework includes datasets spanning three domains—industrial, medical, and general-purpose—and incorporates fifteen state-of-the-art anomaly detection methods along with nine comprehensive evaluation metrics.
3. The experimental results are comprehensive, demonstrating the convenience and versatility of the framework in the field of visual anomaly detection.

**Weaknesses:**

1. The novelty is not at the ICLR level. The primary aim of this paper is to integrate methods, datasets, and evaluation metrics in the field of visual anomaly detection to create a comprehensive modular framework. All the methods, datasets, and evaluation metrics included are based on existing technologies.
2. As the core content of this paper, the details of the ADer framework are not detailed enough. Only Figure 3 briefly describes the core sub-modules of the framework. The main purpose of ADer framework is to integrate the existing VAD algorithms together for convenient and fair comparison and evaluation. However, the process and method of integrating these algorithms are not thoroughly explained, nor are the impacts of integration compared to non-integration detailed or experimentally demonstrated. It is recommended to refer to the comprehensive description of the framework provided in the object detection framework MMDetection [1].
3. As a point mentioned separately, the indicator calculation acceleration algorithm ADEval is not described in detail enough. It is only briefly described in Section 3.6, without any introduction to the iterative-accumulating algorithm it uses. In addition, Section 3.6 only uses an example to describe the improvement of the calculation time of the algorithm, without any supporting experimental data.
4. The primary aim of this paper is to offer a new and fair benchmark for comparing existing visual anomaly detection methods. In the experimental setup, only two parameters—image resolution and training epochs—are fixed, while other parameters remain as defined by the original methods, which does not fully demonstrate ADer’s role and fairness. All experimental results can be compared and verified using the original method's code with the aforementioned parameters standardized.

[1] Kai Chen, Jiaqi Wang, Jiangmiao Pang, et al. MMDetection: Open MMLab Detection Toolbox and Benchmark

**Questions:**

1. Can you provide a detailed explanation of the innovations introduced in this paper for the field of visual anomaly detection? While the integration of existing methods, datasets, and evaluation metrics does contribute to the field's development, it does not constitute an academic research innovation.
2. Can you outline the method and process for integrating existing or self-developed methods into ADer?
3. Could you elaborate on the optimizations ADer brings to existing methods integrated within the framework?
4. Can you provide experimental data from ADEval to support its impact on time-consuming metrics, such as mAU-PRO improvements on large-scale datasets?

---

### Official Review · Reviewer_r81d · 2024-11-02

**Soundness:** 4
**Presentation:** 4
**Contribution:** 3
**Rating:** 8
**Confidence:** 4

**Summary:**

The authors introduce a comprehensive visual anomaly detection benchmark library called ADer to address potential unfair comparisons in the field of multi-class anomaly detection. This library implements 15 state-of-the-art anomaly detection methods on 11 popular datasets with 9 comprehensive evaluation metrics within a scalable framework. The author also develops the ADEval package to tackle the slow evaluation speed of metrics like mAU-PRO on large-scale datasets. Additionally, the author thoroughly reports benchmarked results on multiple datasets and provides the source code.

**Strengths:**

-The author addresses the issue of potentially unfair comparisons due to the lack of unified code evaluation for different methods in the Multi-class Anomaly Detection (AD) setting by proposing the extensible ADer library. This library implements 15 state-of-the-art anomaly detection methods on 11 popular datasets with 9 comprehensive evaluation metrics.
- The developed ADEval GPU acceleration package significantly improves evaluation speed for large-scale datasets.
- Comprehensive benchmark results facilitate intuitive comparison of the effectiveness of different methods.
- The author provides the source code for ADer and commits to open-sourcing it, which is expected to become one of the standard open-source libraries for AD. Additionally, we found that this library also supports Single-class settings and Semantic AD datasets such as CIFAR-10.
- The paper is well-organized and the writing is very clear.

**Weaknesses:**

- Reproducibility of experimental errors: The benchmark results are based on single-run outcomes without calculating the standard deviation from repeated runs. What is the rationale behind this approach? Providing an analysis of randomness and reproducibility would enhance the credibility of the paper.
- The meaning of evaluation metrics should be explained in detail.
- Support for Zero-/Few-shot Anomaly Detection (AD) tasks: The authors primarily focus on multi-class AD. Can the proposed method support the recently popular Zero-/Few-shot AD models, such as WinCLIP? This would further expand the applicability of the AD system.
- Some values are missing in Table 6.
- It would be great if conventional class-separate UAD setting (one-class-one-model) is also benchmarked using this framework.

- Minor issues:
  - There are broken links in Line 374 and Line 430.
  - The text in Fig. 1 is too small.

**Questions:**

Please refer to the Weaknesses.

---

### Official Review · Reviewer_MVfk · 2024-11-03

**Soundness:** 3
**Presentation:** 3
**Contribution:** 3
**Rating:** 5
**Confidence:** 4

**Summary:**

This paper introduces ADer, a comprehensive, standardized evaluation protocol and codebase for multi-class anomaly detection (AD). It consolidates methods, datasets, and evaluation metrics in a single code library, and optimizes computational efficiency for calculating metrics like AUROC, mAP, and AUPRO through GPU acceleration (ADEval).

**Strengths:**

The focus on establishing a standardized evaluation protocol is both timely and relevant, addressing a critical gap in the AD field. The authors' provision of an integrated experimental setup offers a valuable foundation for future AD research. The paper also presents compelling experimental results, including analyses of stability, cross-domain dataset correlations, and more.

**Weaknesses:**

1. Alignment with ICLR Scope: While ADer provides a valuable standardized framework, the paper’s relevance to ICLR could be strengthened. Including a discussion of representation learning techniques in AD, or integrating such techniques into the benchmark, could better align the work with ICLR’s focus. Exploring representation learning might also broaden the utility of ADer by connecting it to core themes in machine learning and anomaly detection research.

2. Justification for Multi-class Focus: The paper currently lacks sufficient justification for focusing solely on the multi-class setting. Addressing the rationale behind this choice would improve the discussion. Additionally, several challenges for multi-class AD mentioned in Section 2.2 remain underexplored within ADer. Greater emphasis on how ADer tackles these issues could clarify its contributions to multi-class AD.

3. Paper Structure: Section 3.5 could be more effectively placed within Section 2 to better showcase ADer’s strengths over prior works. This repositioning would improve the flow and help readers appreciate ADer’s advantages earlier in the paper.

**Questions:**

1. Could you elaborate on how ADer could incorporate representation learning elements to strengthen its relevance to the ICLR audience?
2. What specific challenges in multi-class AD, noted in Section 2.2, were intended to be addressed by ADer, and how does the tool approach these?

---

### Note · Authors · 2024-11-16

I have read and agree with the venue's withdrawal policy on behalf of myself and my co-authors.